# Developmental alterations in centrosome integrity contribute to the post-mitotic state of mammalian cardiomyocytes

David C Zebrowski[1,2]*, Silvia Vergarajauregui[1], Chi-Chung Wu[3], Tanja Piatkowski[2], Robert Becker[1], Marina Leone[1,2], Sofia Hirth[4], Filomena Ricciardi[5], Nathalie Falk[6], Andreas Giessl[6], Steffen Just[4], Thomas Braun[2], Gilbert Weidinger[3], Felix B Engel[1,2]*

[1]Experimental Renal and Cardiovascular Research, Department of Nephropathology, Institute of Pathology, Friedrich-Alexander-Universität Erlangen-Nürnberg, Erlangen, Germany; [2]Department of Cardiac Development and Remodeling, Max Planck Institute for Heart and Lung Research, Bad Nauheim, Germany; [3]Institute for Biochemistry and Molecular Biology, University of Ulm, Ulm, Germany; [4]Department of Medicine II, University of Ulm, Ulm, Germany; [5]Department of Developmental Genetics, Max Planck Institute for Heart and Lung Research, Bad Nauheim, Germany; [6]Department of Biology, Animal Physiology, Friedrich-Alexander-Universität Erlangen-Nürnberg, Erlangen, Germany

*For correspondence: david.
zebrowski@gmail.com (DCZ);
felix.engel@uk-erlangen.de (FBE)

Competing interests: The
authors declare that no
competing interests exist.

Reviewing editor: Yukiko M
Yamashita, University of
Michigan, United States

**Abstract** Mammalian cardiomyocytes become post-mitotic shortly after birth. Understanding how this occurs is highly relevant to cardiac regenerative therapy. Yet, how cardiomyocytes achieve and maintain a post-mitotic state is unknown. Here, we show that cardiomyocyte centrosome integrity is lost shortly after birth. This is coupled with relocalization of various centrosome proteins to the nuclear envelope. Consequently, postnatal cardiomyocytes are unable to undergo ciliogenesis and the nuclear envelope adopts the function as cellular microtubule organizing center. Loss of centrosome integrity is associated with, and can promote, cardiomyocyte G0/G1 cell cycle arrest suggesting that centrosome disassembly is developmentally utilized to achieve the post-mitotic state in mammalian cardiomyocytes. Adult cardiomyocytes of zebrafish and newt, which are able to proliferate, maintain centrosome integrity. Collectively, our data provide a novel mechanism underlying the post-mitotic state of mammalian cardiomyocytes as well as a potential explanation for why zebrafish and newts, but not mammals, can regenerate their heart.

## Introduction

The adult mammalian heart is considered to be a post-mitotic organ, as mammalian cardiomyocytes lose their ability to proliferate shortly after birth (*Li et al., 1996*; *Soonpaa et al., 1996*; *Zebrowski and Engel, 2013*). This is supported by the limited regenerative capacity of the adult mammalian heart (*Senyo et al., 2014*) and the fact that primary adult cardiomyocyte-born tumors are extremely rare, if they exist at all (*Dell'Amore et al., 2011*). Understanding the underlying mechanisms governing the post-mitotic state of adult mammalian cardiomyocytes may clarify whether it is possible to induce cardiac regeneration based on cardiomyocyte proliferation as seen in zebrafish and newts (*Poss et al., 2002*; *Bettencourt-Dias et al., 2003*; *Gamba et al., 2014*). Tremendous efforts have been invested to induce postnatal mammalian cardiomyocyte proliferation. However, besides the fact that cell cycle promoting factors (e.g., Cyclins) are downregulated around birth, while cell cycle inhibitors

**eLife digest** Muscle cells in the heart contract in regular rhythms to pump blood around the body. In humans, rats and other mammals, the vast majority of heart muscle cells lose the ability to divide shortly after birth. Therefore, the heart is unable to replace cells that are lost over the life of the individual, for example, during a heart attack. If too many of these cells are lost, the heart will be unable to pump effectively, which can lead to heart failure. Currently, the only treatment option in humans with heart failure is to perform a heart transplant.

Some animals, such as newts and zebrafish, are able to replace lost heart muscle cells throughout their lifetimes. Thus, these species are able to fully regenerate their hearts even after 20% has been removed. This suggests that it might be possible to manipulate human heart muscle cells to make them divide and regenerate the heart. Recent research has suggested that structures called centrosomes, known to be required to separate copies of the DNA during cell division, are used as a hub to integrate the initial signals that determine whether a cell should divide or not.

Here, Zebrowski et al. studied the centrosomes of heart muscle cells in rats, newts and zebrafish. The experiments show that the centrosomes in rat heart muscle cells are dissembled shortly after birth. Centrosomes are made of several proteins and, in the rat cells, these proteins moved to the membrane that surrounded the nucleus. On the other hand, the centrosomes in the heart muscle cells of the adult newts and zebrafish remained intact.

Further experiments found that that breaking apart the centrosomes of heart muscle cells taken from newborn rats stops these cells from dividing. Zebrowski et al.'s findings suggest that the loss of centrosomes after birth is a possible reason why the hearts of adult humans and other mammals are unable to regenerate after injury. In the future, these findings may aid the development of methods to regenerate human heart muscle and new treatments that may limit division of cancer cells.

(e.g., Cyclin-dependent kinase inhibitors) are induced (*Ikenishi et al., 2012*), little is known about the mechanisms that induce cell cycle exit or establish the post-mitotic state in mammalian cardiomyocytes (*van Amerongen and Engel, 2008*).

The centrosome is a solitary, juxtanuclear organelle in metazoan cell-types. It consists of a pair of tubulin-based structures, called centrioles, encased in a dense, non-membranous, multi-protein cloud called the pericentriolar matrix (PCM), which is further surrounded by a dispersed array of proteins termed centriolar satellites (*Bettencourt-Dias, 2013*). Traditionally, the centrosome is known as the microtubule organizing center (MTOC) of the cell, and is required for primary cilium formation (*Kim and Dynlacht, 2013*). Recently, the centrosome has emerged as a critical signaling hub for the cell cycle regulatory machinery (*Doxsey et al., 2005*). For instance, centrosome localization of Cyclin E and Cyclin A is required for G1/S cell cycle progression (*Pascreau et al., 2011*). Consistent with this role, increasing evidence supports the requirement of a functional centrosome for proliferative potential in mammalian cell-types (*Doxsey et al., 2005*). For instance, cells undergo G0/G1 arrest when (i) centrosomes are removed via laser ablation, (ii) centrosome integrity is disrupted via knockdown of centrosome proteins, or (iii) centriole biogenesis is blocked via a chemical inhibitor (*Hinchcliffe et al., 2001*; *Khodjakov and Rieder, 2001*; *Srsen et al., 2006*; *Mikule et al., 2007*; *Wong et al., 2015*).

To date, the role of centrosome integrity in cell proliferation has always been studied in the context of centrosome component mutants. Here, we show that centrosome integrity is developmentally regulated in mammalian cardiomyocytes, revealing a novel mechanism that renders cells post-mitotic. Our findings might have important implications for efforts to induce therapeutic cardiomyocyte proliferation in adult mammalian hearts.

## Results

A normal diploid cell in G0/G1-phase contains one centrosome with two paired centrioles. During S-phase the centrosome duplicates whereby the two parental centrioles form daughter centrioles. Around the transition from G2-phase to mitosis, parental centrioles 'split' (loss of cohesion) resulting in two separated centrosomes that become part of the spindle poles during mitosis (*Doxsey et al., 2005*). When centrosome integrity is compromised during G0/G1-phase, centrioles can lose cohesion

(i.e., become unpaired) adopting a premature 'split' configuration, defined as a distance > 2 µm between centrioles (*Graser et al., 2007*). To determine if centrosome integrity changes in cardiomyocytes during development, we analyzed centriole configuration using antibodies against γ-tubulin, a marker of both the centriole and the PCM (*Sonnen et al., 2012*). Centrioles were observed to be in a typical paired configuration in the majority of cultured cardiomyocytes isolated from embryonic day (E) 15, E18, and postnatal day (P) 0 rat hearts (*Figure 1A,B*) as well as in cardiomyocytes from E15 and P0 rat heart sections (*Figure 1—figure supplement 1A,B*). In contrast, shortly after birth, centrioles were split in the vast majority of P3 and P5 cardiomyocytes in vitro (*Figure 1A,B*) and in vivo (*Figure 1—figure supplement 1A,B*). Analysis of mother and daughter centriole markers (Odf2 and Centrobin, respectively) in isolated rat cardiomyocytes verified that each γ-tubulin signal represented a single centriole (*Figure 1—figure supplement 1C*). As an internal control, cardiac non-myocytes from the same cultures and tissue sections were examined. The majority of non-myocytes in the heart are fibroblasts, endothelial cells, and smooth muscle cells that all have the ability to proliferate. At all examined developmental stages, non-myocytes showed a typical paired-centriole configuration indicating that the split-centriole phenotype was cardiomyocyte-specific (*Figure 1B* and *Figure 1—figure supplement 1A,B*). Collectively, these data demonstrate that centriole cohesion, and thus centrosome integrity, is lost in mammalian cardiomyocytes shortly after birth.

To identify an underlying cause of the split-centriole phenotype, the cellular localization of various centrosome proteins was assessed in isolated cardiomyocytes from different developmental stages. The PCM proteins Pericentrin and Cdk5Rap2 have previously been shown to be required for centriole-cohesion (*Graser et al., 2007*; *Matsuo et al., 2010*). Consistent with this function, both PCM proteins localized to the centrosome in E15-isolated cardiomyocytes (*Figure 1C,D* and *Figure 1—figure supplement 1D,E*). In contrast, both proteins were localized to the nuclear envelope in P3-isolated cardiomyocytes (*Figure 1C,D* and *Figure 1—figure supplement 1D,E*). Although remnants of Pericentrin and Cdk5Rap2 could be observed at the centriole in P3-isolated cardiomyocytes (*Figure 1C,D* and *Figure 1—figure supplement 1D,E*), their presence was significantly reduced when centrioles were split (*Figure 1E* and *Figure 1—figure supplement 1F*). *Pcnt* siRNA-mediated knockdown in P0-isolated cardiomyocytes resulted in an increase of split-centrioles (*Figure 1F,G*), confirming that Pericentrin is required for centriole-cohesion in cardiomyocytes. In contrast to PCM proteins, the centriole-associated proteins CEP135, Odf2, and Centrobin were not observed at the nuclear envelope in P3-isolated cardiomyocytes (*Figure 1C* and *Figure 1—figure supplement 1C,D*). Collectively, these results indicate that loss of centriole cohesion is accompanied by redistribution of centrosome proteins to the nuclear envelope.

Centrosome localization of Pericentrin is dependent on the centriole satellite protein PCM1 (*Dammermann and Merdes, 2002*; *Barenz et al., 2011*). Consistent with this, PCM1 localized at the centrosome and the nuclear envelope in E15- and P3-isolated cardiomyocytes, respectively (*Figure 1C,H* and *Figure 1—figure supplement 1D*). Further, nuclear envelope localization of PCM1 occurred by P0 (*Figure 1H*), prior to that of Pericentrin (*Figure 1D*). Occasionally, PCM1 was observed in a semi-belt pattern at the nuclear envelope proximal to the centrosome in E15-isolated cardiomyocytes (*Figure 1—figure supplement 1G*), indicating a transitional state. These data suggest that in cardiomyocytes, PCM1 is also required for the localization of Pericentrin to the centrosome. To test this, we overexpressed the major isoform of Pericentrin associated with the centrosome, Pericentrin B, in E15-, P0-, and P3-isolated cardiomyocytes and non-myocytes. Pericentrin B-GFP localized to centrioles in non-myocytes as well as E15-isolated cardiomyocytes (*Figure 1—figure supplement 2A,B*). In contrast, Pericentrin B-GFP did not localize to the centrioles in the vast majority of P0- and P3-isolated cardiomyocytes but rather created cellular aggregates which were not observed in E15 cardiomyocytes or non-myocytes. These data indicate that the machinery required for centrosome integrity is lost in postnatal cardiomyocytes.

To determine if relocalization of centrosome proteins to the nuclear envelope and loss of centriole-cohesion is caused by cell autonomous mechanisms, long-term culturing experiments were performed. Long-term culturing of E15-isolated and P0-isolated cardiomyocytes resulted in PCM1 and Pericentrin relocalization to the nuclear envelope (*Figure 1I*) and loss of paired-centrioles (*Figure 1J*), respectively. Further, mouse iPSC-derived cardiomyocytes had split-centrioles and the centrosome proteins PCM1 and Pericentrin localized to the nuclear envelope (*Figure 1—figure supplement 2C*). These results indicate that loss of centrosome integrity begins during fetal development and progresses in a cell autonomous manner.

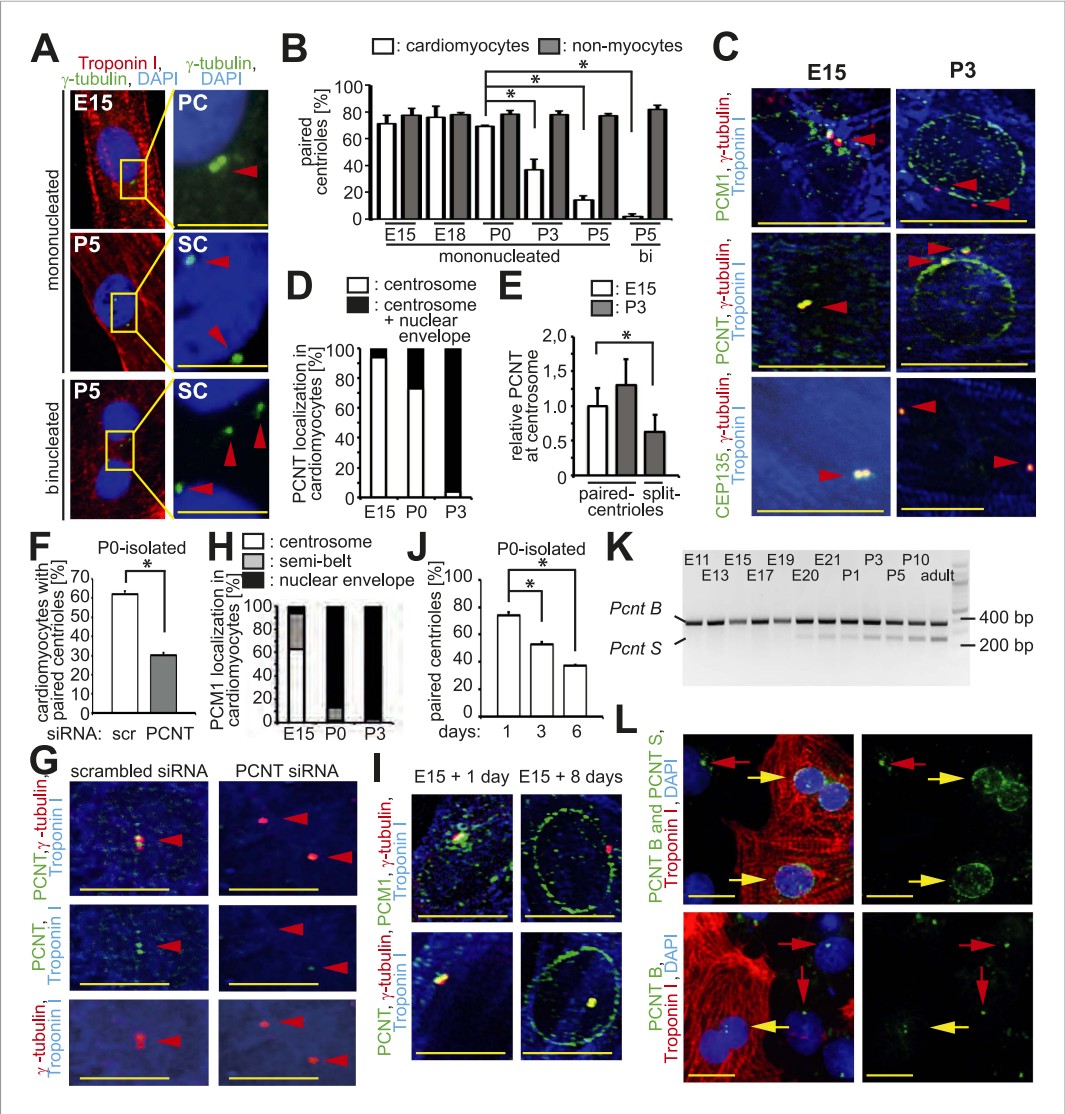

**Figure 1**. Loss of centrosome integrity during heart development. (**A**) Analysis of centriole (γ-tubulin) configuration in E15- or P5-isolated ventricular rat cardiomyocytes (Troponin I). Nuclei: DAPI. PC: paired-centrioles. SC: split-centrioles. Scale bar: 5 μm. (**B**) Frequency of cells with paired-centrioles during development. Bi: Binucleated. (**C**) Analysis of the localization of the centrosome proteins PCM1, PCNT (Pericentrin), and CEP135 in isolated cardiomyocytes. (**D**) PCNT localization frequency in cardiomyocytes isolated from different developmental stages. (**E**) Centrosomal PCNT signal intensity in P3-isolated cardiomyocytes with paired- and split-centrioles relative to E15-isolated cardiomyocytes. (**F**) Frequency of paired-centrioles in P0-isolated cardiomyocytes after siRNA-mediated *Pcnt* knockdown. scr: scrambled. (**G**) Representative images of the analysis in (**F**). (**H**) PCM1 localization frequency in cardiomyocytes isolated from different developmental stages. (**I**) Analysis of PCM1 and PCNT localization in E15-isolated cardiomyocytes cultured for either 1 or 8 days. (**J**) Frequency of P0-isolated cardiomyocytes with paired-centrioles after 1 day, 3 days, or 6 days in culture. (**K**) RT-PCR analysis of *Pcnt B* and *S* isoform expression during rat heart development in vivo. (**L**) Localization of PCNT isoforms. P3-isolated cardiomyocytes immunostained with antibodies against either both PCNT B and S isoforms or only the PCNT B isoform. Yellow arrows: cardiomyocyte nuclei. Red arrows: non-myocyte nuclei. Unless otherwise noted, scale bars: 10 μm; red arrowheads: centrioles; data are mean ± SD, n = 3, *: p < 0.05. For the experiments ≥ 10 cells (**E**), ≥ 50 cells (**B**, **F**, **J**), ≥ 100 (**D**, **H**) cells were analyzed per experimental condition.

The following figure supplements are available for figure 1:

**Figure supplement 1**. Loss of centrosome integrity during heart development.

**Figure supplement 2**. Loss of centrosome integrity during heart development.

Previously, it has been demonstrated that *Pcnt* is alternatively spliced (*Miyoshi et al., 2006*), resulting in two isoforms; Pericentrin B which resembles the human Pericentrin Kendrin, and Pericentrin S, which lacks an N-terminus region. Pericentrin B is ubiquitously expressed at all developmental stages. In contrast, expression of Pericentrin S starts late in fetal development and was found to be specific for adult heart and skeletal muscle (*Miyoshi et al., 2006*). Therefore, we speculated that expression of the Pericentrin S isoform may coincide with changes in centrosome integrity during the development of cardiomyocytes. RT-PCR analysis revealed the appearance of *Pcnt S* expression shortly before birth in the heart (*Figure 1K*). Antibodies specific for the Pericentrin B isoform indicated that Pericentrin S, and not Pericentrin B, was the predominant Pericentrin isoform at the nuclear envelope in P3-isolated cardiomyocytes (*Figure 1L*). These results indicate that *Pcnt* is alternatively spliced during perinatal heart development and this change is related to its relocalization to the nuclear envelope.

Loss of centrosome integrity in postnatal cardiomyocytes suggested that centrosome function might be compromised as well. Nearly all cell-types studied to date are capable of forming a primary cilium when arrested in G0/G1-phase (*Bowser and Wheatley, 2000*, *Nigg and Stearns, 2011*). This suggests that cardiomyocytes, which are arrested shortly after birth in G0/G1 phase (*Takeuchi, 2014*), should be capable of ciliogenesis. Yet, given that PCM1, which is required for ciliogenesis (*Kim et al., 2008*), is lost from the centrosome during development, we speculated that ciliogenesis is suppressed in cardiomyocytes. Induction of ciliogenesis via serum starvation resulted in the formation of a primary cilium in E15-isolated cardiomyocytes and non-myocytes (*Figure 2A*). However, the frequency of cardiomyocytes capable of ciliogenesis decreased with heart development (*Figure 2B*) with less than 1% of isolated postnatal cardiomyocytes, which were arrested in G1/G0 as confirmed by lack of Ki67 expression and FACS analysis (*Figure 2—figure supplement 1A–D*), forming a primary cilium. Further, no postnatal binucleated cardiomyocytes were observed to be capable of ciliogenesis (*Figure 2A*). In contrast to cardiomyocytes, the frequency of cardiac non-myocytes capable of ciliogenesis was high at all developmental stages investigated (*Figure 2B*).

Pericentrin and Cdk5Rap2 are required for the centrosome to function as the cellular MTOC (*Takahashi et al., 2002*; *Choi et al., 2010*). As these proteins localize to the nuclear envelope during neonatal development, we hypothesized that the cellular MTOC is transferred from the centrosome to the nuclear envelope. In accordance with this hypothesis, microtubules were found to predominantly emanate from the centrosome in E15-isolated cardiomyocytes as also observed in non-myocytes. In contrast, in P3-isolated cardiomyocytes microtubules were found to predominantly emanate from the nuclear envelope (*Figure 2C*). The local shift of the MTOC during development was confirmed by a MTOC-regrowth assay (*Figure 2D* and *Figure 2—figure supplement 1E,F*), which further demonstrated that microtubules do not emanate from centrioles in P3-isolated cardiomyocytes (*Figure 2E* and *Figure 2—figure supplement 1G,H*). Finally, siRNA-mediated knockdown of *Pcnt* in P3-isolated cardiomyocytes demonstrated that Pericentrin is required for a functional MTOC at the nuclear envelope (*Figure 2F* and *Figure 2—figure supplement 1I*). Collectively, these results indicate that, in addition to centrosome integrity, centrosome function is progressively compromised in cardiomyocytes during development.

Given that traditional centrosome functions of ciliogenesis and microtubule organization are lost in postnatal cardiomyocytes, we hypothesized that there would be a relationship between centrosome integrity and proliferative potential. To test this, the cell cycle marker Ki67 was assessed in cardiomyocytes from different developmental stages. Cardiomyocyte proliferative potential decreased with neonatal development (*Figure 3A*). Moreover, P3-isolated cardiomyocytes with paired-centrioles exhibited greater proliferative potential than those with split centrioles (*Figure 3B,C*) at a frequency similar to that of E15- and P0-isolated cardiomyocytes (*Figure 3A,C*)—of which the vast majority have paired-centrioles (*Figure 1B*). In contrast, cardiac non-myocyte proliferative potential, as well as the percentage of non-myocytes with paired centrioles, did not decrease during neonatal development (*Figure 3—figure supplement 1A*). Subsequently, we tested whether centrosome integrity is required for cardiomyocyte proliferative potential. Centrosome integrity was disrupted by either siRNA-mediated knockdown of *Pcnt*, or overexpression of a RFP-tagged dominant negative C-terminal Pericentrin (RFP-Peri[CT]), which displaces endogenous Pericentrin localization (*Gillingham and Munro, 2000*; *Mikule et al., 2007*). Immunofluorescence analysis confirmed that RFP-Peri[CT] localizes to the centrioles in P0-isolated cardiomyocytes (*Figure 3—figure supplement 1B*). Both methods used to disrupt centrosome integrity suppressed P0-isolated cardiomyocyte proliferative

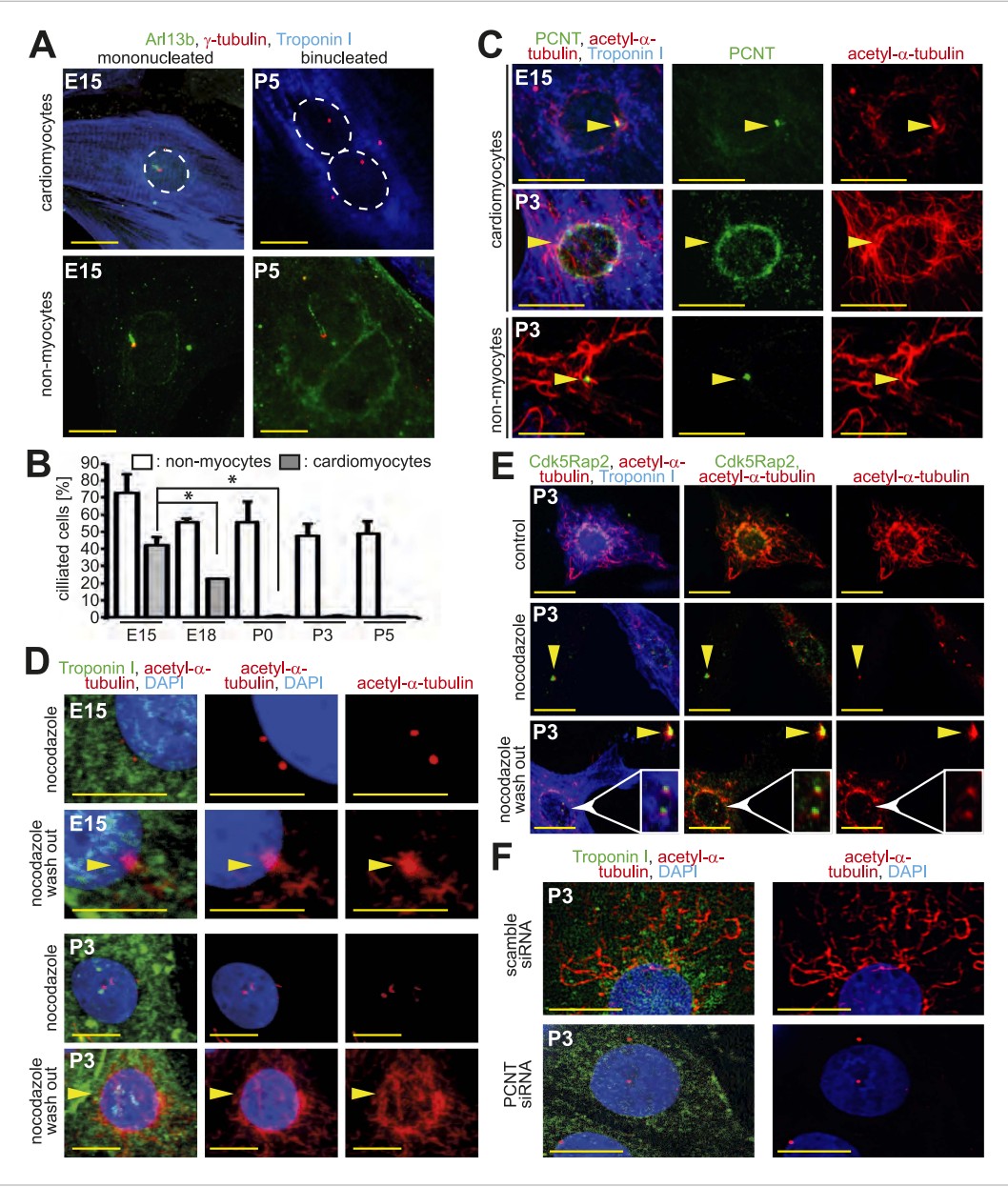

**Figure 2.** Loss of centrosome function during heart development. (**A**) Identification of primary cilium (Arl13b) in E15- and P5-isolated cardiomyocytes (Troponin I) and non-myocytes. Centrioles: γ-tubulin. White circle: nuclei. (**B**) Frequency of ciliated cardiomyocytes and non-myocytes isolated from hearts at different developmental stages. Data are mean ± SD, n = 3, *: p < 0.05. ≥ 200 cells were analyzed for each condition. (**C**) Localization of the cellular MTOC (Pericentrin [PCNT]) in E15- and P3-isolated cardiomyocytes. Microtubules: acetylated-α-tubulin; yellow arrowheads: PCNT-positive MTOC. (**D**) Localization of microtubule regrowth (yellow arrowhead). E15- or P3-isolated cardiomyocytes treated with nocodazole or nocodazole followed by wash-out. (**E**) Analysis of microtubule regrowth at centrioles. P3-isolated cardiomyocytes were treated as in (**D**). Yellow arrowheads: Cdk5Rap2-positive centrioles in non-myocytes; White arrowheads: Cdk5Rap2-positive centrioles in cardiomyocytes. (**F**) Pericentrin is required for a functional MTOC. P3-isolated cardiomyocytes transfected with scrambled or PCNT siRNAs and analyzed for microtubule formation. Scale bars: 10 μm.

The following figure supplement is available for figure 2:

**Figure supplement 1**. Loss of centrosome function during heart development.

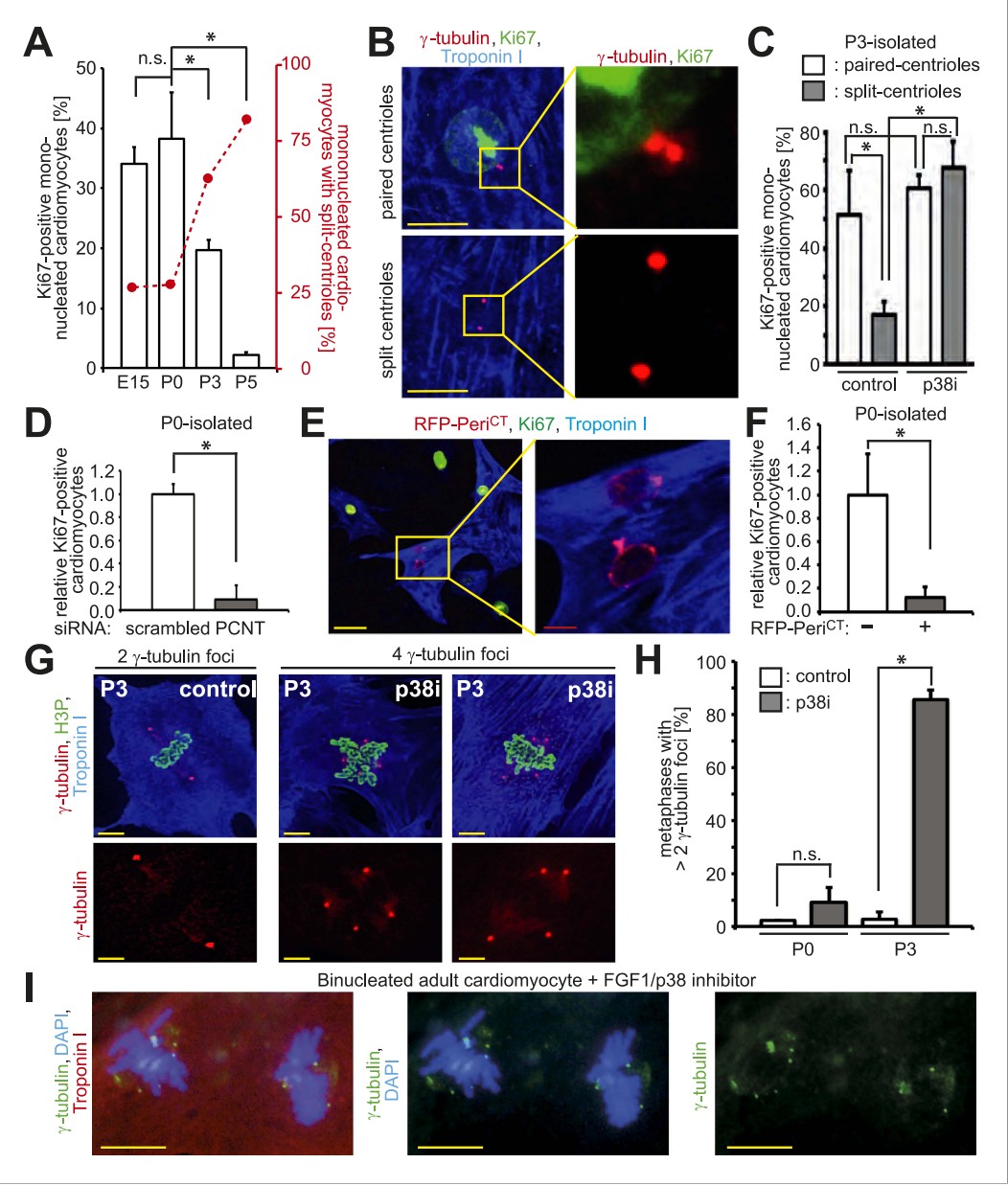

**Figure 3**. Absence of centrosome integrity results in cell cycle aberrations. (**A**) Correlation between proliferative potential (Ki67) and centrosome integrity (split-centrioles). Overlay of the frequency of Ki67-positive E15-, P0-, P3-, or P5-isolated cardiomyocytes in response to 20% fetal bovine serum (FBS) (bars) and the frequency of cardiomyocytes with split-centrioles at different developmental stages according to *Figure 1B* (red line). (**B**) Representative images indicating that 20% FBS-stimulated P3-isolated cardiomyocytes (Troponin I) with paired-centrioles (γ-tubulin) exhibited greater proliferative potential (Ki67) than those with split centrioles. Scale bars: 10 μm. (**C**) Frequency of Ki67-positive P3-isolated cardiomyocytes in response to 20% FBS in the presence or absence of p38 MAP kinase inhibitor (p38i). (**D–F**) Pericentrin (PCNT) is required for cardiomyocyte proliferative potential. (**D**) Relative frequency of Ki67-positive P0-isolated cardiomyocytes treated with scrambled or PCNT siRNA in response to 20% FBS. (**E**) Representative images of 20% FBS-stimulated P0-isolated cardiomyocytes transfected with a construct driving RFP-tagged dominant negative C-terminal Pericentrin (RFP–PeriCT) expression and stained for Ki67. Scale bars: 10 μm. (**F**) Quantitative analysis of (**E**). (**G, H**) Analysis of centrioles (γ-tubulin) during metaphase (H3P). (**G**) Representative images of P3-isolated cardiomyocytes in metaphase in the presence or absence of p38i upon stimulation with 10% FBS. Scale bars: yellow: 50 μm; red: 10 μm. (**H**) Quantitative analysis of (**G**). (**I**) Representative images of centrioles in adult cardiomyocytes in metaphase stimulated with

*Figure 3. continued on next page*

*Figure 3. Continued*

FGF1 plus p38i. Chromosomes: DAPI. Scale bars: 20 μm. Data are mean ± SD, n = 3, *: p < 0.05. For the experiments ≥ 20 cells (**F**), ≥ 25 cells (**D**), ≥ 40 cells (**H**), or ≥ 50 cells (**A**, **C**) were analyzed per experimental condition.

The following figure supplement is available for figure 3:

**Figure supplement 1**. Absence of centrosome integrity results in cell cycle aberrations.

potential (*Figure 3D–F*). Taken together, these data indicate that loss of centrosome integrity promotes G0/G1 cell cycle arrest in mammalian cardiomyocytes.

Disruption of centrosome integrity promotes p38MAP kinase (p38)-mediated G1/S cell cycle arrest (*Mikule et al., 2007*; *Pascreau et al., 2011*). Previously, it has been demonstrated that p38 inhibition (p38i) allows postnatal cardiomyocyte proliferation (*Engel et al., 2005*). This raised the question whether p38i can over-ride post-mitotic arrest in cardiomyocytes with split-centrioles. p38i enhanced the proliferative potential of P3-isolated cardiomyocytes with split-centrioles (*Figure 3C*). These results indicate that suppression of the p38-mediated stress-activated pathway can promote cell cycle progression in cardiomyocytes that lack centriole-cohesion.

A bipolar mitotic spindle is required for high fidelity chromosome segregation (*Pihan, 2013*). Thus, we determined if the observed changes in centrosome integrity affects spindle pole number during mitosis in postnatal cardiomyocytes when cell cycle progression is induced. In response to serum stimulation, P0- and P3-isolated cardiomyocytes exhibited a typical metaphase consisting of two γ-tubulin foci (*Figure 3G,H*). p38i had no effect on the number of γ-tubulin foci in P0-isolated cardiomyocytes during metaphase (*Figure 3H*). In contrast, p38i resulted in a significant increase in metaphases containing multiple γ-tubulin foci in P3-isolated cardiomyocytes (*Figure 3G,H*). Similarly, adult-isolated cardiomyocytes induced to re-enter the cell cycle by p38i + FGF1 also exhibited multiple γ-tubulin foci during metaphase (*Figure 3I*). These data indicate that over-riding cell cycle arrest in cardiomyocytes results in multiple spindle poles.

In contrast to mammals, adult newts and zebrafish can regenerate their heart through cardiomyocyte proliferation (*Poss et al., 2002*; *Bettencourt-Dias et al., 2003*). This suggests that they either do not establish a post-mitotic state by disassembling their centrosomes or they are able to reverse this mechanism upon injury. To test which scenario occurs in these species, we analyzed their centrosomes. Paired centrioles could readily be identified in adult newt and zebrafish cardiomyocyte nuclei (*Figure 4A*). Further, the frequency of newt and zebrafish cardiomyocyte nuclei with intact centrosomes was similar to that of non-myocyte nuclei (*Figure 4B*). In addition, MTOC-regrowth assays demonstrated that centrosomes in zebrafish and newt cardiomyocytes are functional (*Figure 4C* and *Figure 4—figure supplement 1A,B*). Collectively, these results suggest that, unlike mammalian cardiomyocytes, newt and zebrafish cardiomyocytes maintain centrosome integrity throughout adulthood.

We then sought to determine if zebrafish cells require centrosome integrity for proliferation in vivo. To determine this, we injected RNA encoding RFP-tagged dominant negative C-terminal Pericentrin (RFP-Peri[CT]) into one-cell stage zebrafish embryos. Compared to control embryos (RFP RNA-injected), a significantly larger number of RFP-Peri[CT] RNA-injected embryos showed developmental delay at 24 hr post fertilization (hpf) and a significantly reduced number of H3P-positive cells (*Figure 4—figure supplement 1C–F*). Since RFP-Peri[CT] RNA did not cause necrosis or lethality, it is plausible that the observed developmental delay was caused, at least partly, by reduced proliferation of RFP-Peri[CT]-expressing cells. To further test an effect on proliferation, we injected DNA constructs driving RFP or dominant negative Pericentrin (RFP-Peri[CT]) expression from the CMV promoter into one-cell stage embryos and analyzed the size of RFP-positive clones of cells at 20 hpf. Embryos were sorted into four classes according to the size of the clones, namely no visible clones (class I), few scattered cells (class II), medium-sized clones (class III), and large clones (class IV). RFP DNA-injected controls exhibited comparable numbers of embryos displaying no clones (class 1), scattered (class II), medium-sized (class III), and larger clones (class IV). In contrast, the majority of RFP-Peri[CT] DNA-injected embryos contained either no clones (class I) or few scattered clones (class II), while larger clones (class III and IV) were rarely seen (*Figure 4—figure supplement 1G,H*). This suggests that clonal expansion of RFP-Peri[CT] expressing cells is reduced, which substantiates the hypothesis that expression of RFP-Peri[CT] reduces cell proliferative potential. These results suggest that zebrafish cells require centrosome integrity for proliferation in vivo.

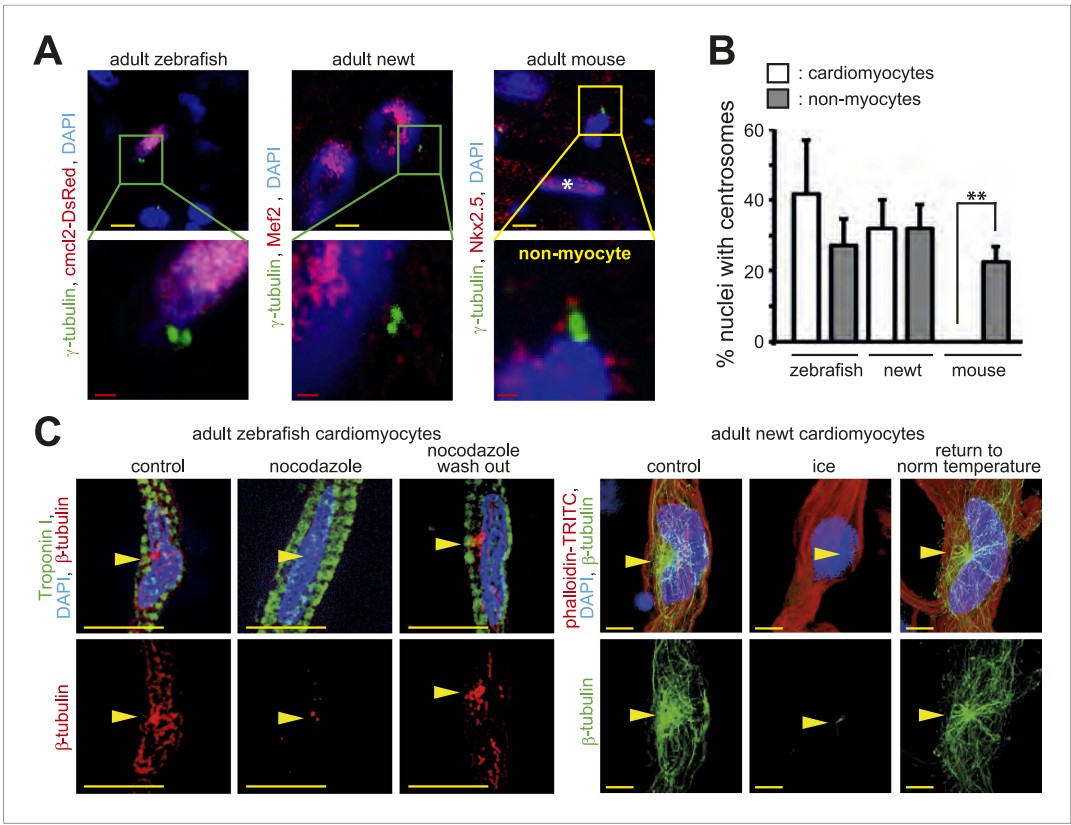

**Figure 4**. Centrosome integrity is maintained in adult newt and zebrafish cardiomyocytes. (**A**) Representative images of centrosomes (γ-tubulin) in heart cryosections of adult transgenic *cmlc2:dsRedExp-nuc^hsc4* zebrafish, of adult newt hearts, and adult mouse hearts. Nuclei: DAPI, cardiac nuclei: DsRed or Nkx2.5. Green-framed expansions: newt and zebrafish cardiomyocyte nuclei with paired-centrioles. Yellow-framed expansion: mouse non-myocyte nucleus with paired-centrioles. White asterisk: cardiomyocyte nucleus. Yellow scale bar: 10 μm. Red scale bar: 2 μm.
(**B**) Quantitative analysis of nuclei associated with intact centrosomes in cryosections as shown in (**A**). Data are mean ± SD, n = 3, ≥ 300 cells were analyzed per experimental condition, *: p < 0.05. (**C**) Representative images documenting localization of microtubule regrowth (β-tubulin) in cultured adult zebrafish (Troponin I) and newt (phalloidin-TRITC) cardiomyocytes. Adult zebrafish were treated with serum (control), nocodazole, or nocodazole followed by wash out. Adult newt cardiomyocytes were treated with serum (control), ice, or ice followed by return to normal (norm) temperature. Nuclei: DAPI. Yellow arrowheads: localization of centrosome. Yellow scale bar: 10 μm.

The following figure supplement is available for figure 4:

**Figure supplement 1**. Centrosome integrity is maintained in adult newt and zebrafish cardiomyocytes.

## Discussion

Increasing evidence supports the requirement of a functional centrosome for cellular proliferative potential (*Doxsey et al., 2005*). For instance, somatic cell-types arrest in G0/G1 when centrosomes are removed or disrupted (*Hinchcliffe et al., 2001*; *Khodjakov and Rieder, 2001*; *Srsen et al., 2006*; *Mikule et al., 2007*; *Wong et al., 2015*). Further, centrosome degradation occurs during female meiosis—a process believed to inhibit parthenogenesis—with mitotic cycles restored upon fertilization when the sperm donates a centrosome during zygote formation (*Klotz et al., 1990*; *Clift and Schuh, 2013*). Moreover, relocalization of the MTOC to non-centrosomal loci has been described for several post-mitotic, highly differentiated, cell-types (*Bartolini and Gundersen, 2006*). This body of literature coupled with our results suggests that centrosome integrity can be developmentally regulated to achieve a post-mitotic state.

Centrosome disassembly appears to be a very effective way to achieve a post-mitotic state. But why do cardiomyocytes disassemble their centrosomes? Upon birth, the neonatal heart, and the

cardiomyocytes therein, undergo increased hemodynamic stress. Effective cardiomyocyte function in response to increased hemodynamic stress may require a cytoskeletal architecture more conducive to handling postnatal physical stresses (e.g., a nuclear envelope-based MTOC). Thus, centrosome disassembly may be a result of cytoskeletal reorganization. In this scenario, proliferative potential might be sacrificed for postnatal function.

It has long been considered that skeletal myoblasts and cardiomyocytes have distinct molecular mechanisms to control their proliferative growth. This theory has been primarily based on the observation that proliferation and differentiation (contraction) are mutually exclusive in skeletal muscle but occur in parallel in cardiomyocytes (*Ueno et al., 1988*). However, our results together with others (*Tassin et al., 1985*; *Bugnard et al., 2005*; *Srsen et al., 2006*; *Fant et al., 2009*; *Zaal et al., 2011*) challenges this dogma, as establishment of a post-mitotic state in both muscle cell-types coincides with MTOC relocalization to the nuclear envelope and centrosome disassembly.

The observation that p38i promotes cardiomyocyte proliferative potential in the absence of a functional centrosome provides some optimism for heart regeneration via proliferation of endogenous cardiomyocytes. However, we observe that an absence of centrosome integrity correlates with multiple spindle poles. Although multiple spindle poles are generally resolved into a semi-clustered/pseudo-bipolar conformation, this process is nevertheless highly prone to the formation of merotelic spindle-kinetochore attachments, which can promote chromosome missegregation (*Ganem et al., 2009*). Indeed, postnatal cardiomyocytes induced to proliferate exhibit chromosome segregation abnormalities (*Engel et al., 2006*). While cell viability can exist when aneuploidy is limited (e.g., as seen in Down Syndrome), aneuploidy is generally not well tolerated (*Torres et al., 2008*). Therefore, proliferation of adult cardiomyocytes may not necessarily result in viable daughter cells. Thus, in the absence of a functional centrosome, whether a particular manipulation induces aneuploidy, and to what degree, may be a critical factor in determining its regenerative therapeutic potential.

The origin of multiple spindle poles (i.e., multiple γ-tubulin foci) during cardiomyocyte mitosis is not entirely clear. Interestingly, in addition to centriole-cohesion, Pericentrin and Cdk5Rap2 are also required for centriole-engagement (*Barrera et al., 2010*; *Lee and Rhee, 2012*). During S-phase, centrioles are duplicated, with each mother centriole forming a daughter centriole, which remains closely attached (i.e., engaged) until mid-anaphase (*Kuriyama and Borisy, 1981*; *Sluder, 2013*). When mother-daughter centrioles are engaged, only the mother centriole exhibits a strong γ-tubulin signal (*Wang et al., 2011*). Thus, one speculation is that when cells that lack centriole-cohesion enter mitosis, they have an increased likelihood of premature loss of centriole-engagement, thus accounting for the four γ-tubulin signals observed at metaphase. However, it has to be considered that induction of mitosis in bi-nucleated cardiomyocytes, which increasingly appear after birth, results in four γ-tubulin signals in metaphase corresponding to four duplicated centrioles.

The ability of zebrafish and newts to regenerate their heart has gained extensive interest in recent years. One major question is what distinguishes mammalian cardiomyocytes from those of zebrafish and newts with regards to their proliferative potential. Our data demonstrate that the state of cellular differentiation of cardiomyocytes from various species is not evolutionary conserved. The fact that adult zebrafish and newt cardiomyocytes maintain their centrosome integrity indicates that factors promoting adult zebrafish cardiomyocyte proliferation might not necessarily induce adult mammalian cardiomyocyte proliferation.

Recently, it has been shown that planarians can develop and regenerate in the absence of centrosomes (*Azimzadeh et al., 2012*). This has questioned the requirement of a centrosome for cell proliferation during development and regeneration. However, there are significant differences between planarians and vertebrates in cell cycle control. For example, planarians express only one single repressive E2F, whereas mammals and zebrafish express several repressive and activating E2Fs. In addition, planarians do not have Cyclin E or Cyclin A homologs. Further, cyclin-dependent kinase CDK2 (to which Cyclin E/A usually binds) is expressed at such low levels in planarians that it is considered functionally dead (*Zhu and Pearson, 2013*). This is important as in mammals centrosome localization of Cyclin E and Cyclin A is required for G1/S cell cycle progression (*Pascreau et al., 2011*). Thus, at least in the case of Cyclin E and Cyclin A, planarians lack centrosome-regulated cell cycle factors. In addition, our data indicate that centrosome integrity is required for proliferation during zebrafish development. This is in agreement with the recent observation that plk4 depletion in zebrafish impairs centriolar biogenesis during development and increases premature cell cycle exit (independent of ciliogenesis defects) resulting in reduced zebrafish size (*Martin et al., 2014*). Other

examples of centriole-associated genes whose depletion causes cell cycle defects resulting in impaired development are *stil* (*Pfaff et al., 2007*; *Vulprecht et al., 2012*; *Sun et al., 2014*) and *cetn2* (*Delaval et al., 2011*). While there is no evidence for the requirement of functional centrosomes in cardiac regeneration in zebrafish, there are data that indirectly suggest that centrosome function might be required for cardiac regeneration. For example it has been shown that inhibition of *mps1* and *plk1*, factors implicated in centrosome assembly and maturation (*Pike and Fisk, 2011*; *Joukov et al., 2014*; *Kong et al., 2014*), impairs cardiac regeneration in zebrafish after apex resection (*Poss et al., 2002*; *Jopling et al., 2010*). Thus, it will be interesting in the future to test if destruction of centrosome integrity will indeed abolish the ability of zebrafish to regenerate their heart.

Taken together, this study suggests that relocalization of the MTOC disrupts centrosome integrity which, in turn, promotes a post-mitotic state in mammalian cardiomyocytes. Given the increasing role of the centrosome in cell cycle control, understanding how centrosome integrity is regulated during development may reveal new mechanisms to regulate cell proliferation with implications for regeneration and cancer.

## Materials and methods

### Animals

The investigation conforms with the Guide for the Care and Use of Laboratory Animals published by the Directive 2010/63/EU of the European Parliament and according to the regulations issued by the Committee for Animal Rights Protection of the State of Hessen (Regierungspraesidium Darmstadt) as well as Baden-Württemberg (Regierungspraesidium Tübingen). Extraction of organs and preparation of primary cell cultures were approved by the local Animal Ethics Committee in accordance to governmental and international guidelines on animal experimentation (protocol TS—5/13 Neph-ropatho; Zebrafish protocol number o.183). Adult (1–2 year) zebrafish (*Danio rerio*) hearts were isolated from the transgenic line *Tg(cmlc2:dsRedExp-nuc$^{hsc4}$)* (*Takeuchi et al., 2011*). Adult newt cardiomyocytes were isolated from red-spotted newts (*Notophthalmus viridescens*, Charles Sullivan, Nashville, TN, USA). Ventricular cardiomyocytes and whole-hearts were obtained from embryonic day 15 (E15), E18, postnatal day 0 (P0), P3, P5, and adult Sprague–Dawley rats (from Charles River Laboratories, Cologne, Germany or own bred).

### Cell culture

Mammalian and zebrafish ventricular cardiomyocytes were isolated as described previously (*Engel et al., 2005*; *Sander et al., 2013*). Newt ventricular cardiomyocytes were isolated using the following procedure. To prevent contaminations, animals were kept 24 hr in advance in a Sulfamerazine bath (5 g/l, Sigma, St. Louis, MO, USA) for disinfection. Organ removal was performed under deep anesthesia, by incubating the animals in a Tricaine solution (1 g/l, Sigma) with pH 7.4 for 15–20 min. After decapitation, ventricles were removed, washed several times with 65% L15 Leibovitz media (Gibco, Grand Island, New York, USA) with antibiotics (2% penicillin/streptomycin and ciprofloxacin (10 µg/ml)) and incubated over night at 25°C. Thereafter, enzymatic digestion followed with a sterile mixture of collagenase (1 mg/ml, Sigma), elastase (0.1 mg/ml, Sigma) and DNase (0.1 mg/ml, Sigma) with glucose (3 mg/ml, Sigma) and BSA (1.5 mg/ml, Sigma) in aPBS (75% PBS) for 6 hr at 27°C. After mechanical dissociation, and several washing step, cells were plated on laminin (15 µg/ml, Sigma) coated 8-well chamber slides (Nunc) and cultured for 5 days with 65% MEM with Glutamaxx (Gibco) containing 10% FCS and antibiotics (2% penicillin/streptomycin and ciprofloxacin [10 µg/ml]) at 25°C with 5% $CO_2$, media change took place once after 3 days. Mouse iPSC-derived cardiomyocytes (Axiogenesis AG, Cologne, Germany) were thawed and plated according to manufacturer's instructions. Mammalian cardiomyocytes were cultured on 1 mg/ml fibronectin (Sigma)-coated glass coverslips. Isolated cardiomyocytes were seeded and cultured in DMEM/F-12, Glutamax™-I (Life Technologies, Darmstadt, Germany) + Penicillin (100 U/ml)/Streptomycin (100 µg/ml) (Pen/Step) (Life Technologies) for 2 days prior to experimentation. To analyze proliferative potential of whole cardiomyocyte populations, cardiomyocytes were cultured with 20% fetal bovine serum (FBS) (Sigma) for 2 days. To analyze proliferative potential between cardiomyocytes with paired and split centrioles, cardiomyocytes were cultured with 20% FBS + 2 mM hydroxyurea (HU) (Sigma), for 2 days. As centriole-cohesion is normally lost at G2/M, G1/S arrest with hydroxyurea prevents misinterpreting normal loss of centriole-cohesion (which occurs in G2) for precocious loss of centriole-cohesion (which

occurs during G1). To analyze mitotic aberrations, neonatal cardiomyocytes were cultured with 10% FBS for 2 days, and adult cardiomyocytes were seeded in 10% horse serum and 20 μM cytosine β-D-arabinofuranoside (Sigma) for 2 days and then stimulated with 50 ng/ml FGF1 (R&D Systems, Abingdon, UK) + p38i (5 μM SB203580, Tocris Biosystems, Bristol, UK) for 2 days as previously described (*Engel et al., 2005*). To analyze microtubule regrowth in mammalian and zebrafish cardiomyocytes, cells were cultured for 2 days and then treated with 5 μg/ml nocodazole (Sigma) for 2.5 hr. Subsequently, cells were washed with nocodazole-free media for 5–10 min to allow microtubule regrowth. To analyze microtubule regrowth in newt cardiomyocytes, cells were cultured for 5 days and then placed on ice for 3 hr. Subsequently, cells were returned to 25°C for 10 min. For siRNA studies cells were cultured for 2 days prior to transfection of siRNAs (Qiagen, Venlo, Limburg, Netherlands) using Lipofectamine RNAiMAX (Life Technologies). Cycling and MTOC regrowth assays utilizing siRNAs were conducted 2 days after siRNA transfection. Plasmids were transfected with Lipofectamine LTX (Life Technologies) on the day of seeding. Cycling assays were conducted 2 days after plasmid transfection. To induce ciliogenesis, cardiomyocytes were seeded and cultured in DMEM GlutaMAX™-I (Gibco) + Pen/Strep for 3 days.

## Cryosections

Hearts from zebrafish, newts, Sprague Dawley rats (Charles River Laboratories), or C57BL/6J mice (Charles River Laboratories) were oriented perpendicularly in relation to their long axis, embedded in an O.C.T. compound tissue-freezing medium, and frozen in liquid nitrogen. Hearts were sectioned with a Leica CM 3000 cryostat (10 μm).

## Immunofluorescence analysis

Cryosections were fixed in 3.7% formalin (Sigma) for 10 min at room temperature (RT). Isolated cells were fixed in either pre-chilled methanol for 5 min at −20°C or 3.7% formalin (Sigma) for 10 min at RT. Immunostaining was performed as described previously (*Engel et al., 2005*) utilizing 3% BSA (Sigma)/ PBS instead of goat-serum as blocking buffer. Formalin-fixed cells were permeabilized prior to antibody staining with 0.2% Triton X-100 (Sigma)/PBS (10 min, RT). Primary antibodies: goat anti-Troponin I (1:250, Abcam, Cambridge, UK), rabbit anti-Troponin I (1:250, Santa Cruz Biotechnology, Heidelberg, Germany), rabbit anti-Cdk5Rap2 (1:500, Millipore, Hessen, Germany), rabbit anti-Pericentrin (1:700, Covance, Princeton, NJ, USA), mouse anti-γ-tubulin (1:500, Santa Cruz Biotechnology), rabbit anti-PCM1 (1:500, Santa Cruz Biotechnology), rabbit anti-Odf2 (1;500, ProteinTech Group, Manchester, UK), rabbit anti-Centrobin (1:500, Sigma), goat anti-Nkx2.5 (1:100, Santa Cruz Biotechnology), rabbit anti-CEP135 (1:500, Abcam), rabbit anti-phospho-histone H3-Serine 10 (1:1000, Santa Cruz Biotechnology), rabbit anti-Mef2 (1:500, Santa Cruz Biotechnology), mouse anti-β-tubulin (KMX) (1:500, Millipore), Phalloidin-TRITC (1:300, Sigma). Rabbit anti-Pericentrin (1:500; MmPeriC1) against both B and S isoforms was produced as previously described (*Mühlhans et al., 2011*). Mouse anti-Pericentrin against the Pericentrin B isoform (1:500; MmPeri N-term clone 7H11 or 8D12) was made against the first 233 amino acids of mouse Percentrin B (AN: NP_032813 or BAF36559). Primary immune complexes were detected with ALEXA 350-, ALEXA 488-, ALEXA 594-, or ALEXA 647-conjugated antibodies (1:500, Life Technologies, Carlsbad, CA, USA). DNA was stained with 0.5 μg/ml DAPI (4′,6′-diamidino-2-phenylindole) (Sigma). Images were captured on a Keyence BZ9000 Fluorescence Microscope (Keyence, Osaka, Japan), using 63× or 100× objectives. Images were arranged with ImageJ (Public Domain) and Adobe Illustrator (Adobe, San Jose, CA, USA).

## Quantitation of fluorescence intensity by linescan analysis

For the quantitative analysis of protein intensity in cardiomyocytes and non-myocytes at centriolar loci, a line-plot was generated using software (Keyence) which traversed the centrioles (identified by γ-tubulin staining) and the average Cdk5Rap2 or Pericentrin background intensity was subtracted from signals corresponding to γ-tubulin loci. Cardiomyocyte Cdk5Rap2 or Pericentrin signal intensity was normalized to that of non-myocytes from the same cultures.

## Reverse transcriptase PCR (RT-PCR)

RNA was isolated from different developmental stages of rat (E11 to E20, n ≥ 10; P5, P10, and adult, n ≥ 3) using TRIzol (Life Technologies). RT-PCR was performed following standard protocols. A set of

three primers was used to detect *Pcnt B* and *Pcnt S* in the same reaction. Primers used were Pcnt B-F, 5′-CATGGCTCTGCACAATGAAG-3′, Pcnt S-F 5′-CAGGGCTGTTCCATATGTTC-3′, Pcnt-R 5′-GAAG TCTCCTCAGGGCATCTC-3′.

## FACS analysis

Neonatal cardiomyocytes were cultured for 3 days after isolation. On the third day the cells were washed in PBS, trypsinized, fixed in ice-cold 70% EtOH/15% PBS, and centrifuged (10 min, 700×$g$, 4°C). The cell pellet was resuspended in PBS and centrifuged again. Cells were then resuspended in extraction buffer (50 mM $Na_2HPO_4$: 25 mM citric acid (9:1), 0.1% Triton X-100, 0.01% $NaN_3$, pH 7.8) and incubated for 15 min at RT. Cells were centrifuged and the cell pellet was incubated in 250 µl of complete DNA staining buffer (10 mM PIPES, 0.1 M NaCl, 2 mM $MgCl_2$, 0.1% Triton X-100, 0.02% $NaN_3$, pH 6.8), 15 µl of RNase A (10 mg/ml) and 12 µl of propidium iodide (1 mg/ml) for 30 min at RT. Afterwards the cell suspension was transferred to a FACS tube and 150 µl of PBS was added. Per sample 10,000 events were analyzed with a BD FACSCanto II (BD Transduction, Heidelberg, Baden-Württemberg, Germany) and analysed with the FlowJo software.

## Zebrafish microinjection

RFP-tagged dominant negative C-terminal *Pcnt* was PCR-amplified from the RFP-Peri[CT] construct, using specific primers (forward: 5′gggcccgaattcGCAAACATGGTGACGTCACCGGTCGCCACCATG3′, reverse 5′ gggcccctcgagTCATCGGGTGGCAGGATTTCTTTGAAG 3′) to introduce an EcoRI site at the 5′ end and an XhoI site at the 3′ end and cloned into the EcoRI and XhoI sites of the pCS2+ vector. Subsequently, mCherry (Clontech, Saint-Germain-en-Laye, France) was introduced into the EcoRI and BglII sites replacing mDsRed. Capped sense RNA was synthesized in vitro using mMessage mMachine kits (Life Technologies). RNA (90 pg) or DNA (25 pg) was injected into the cytoplasm of one-cell stage zebrafish embryos using standard procedures. Embryos were raised at 28.5°C until indicated time and were checked for RFP expression.

## Quantification of mitosis in zebrafish embryos

Embryos were raised at 28.5°C until 24 hpf and were fixed in 4% PFA in PBS for 2 hr at RT. After 3 washes in PBST (PBS + 0.1% Tween), embryos were washed in water and were permeabilized in prechilled acetone at −20°C before immunostaining. Mitotic cells were identified using anti-H3P antibodies (1:200, Cell-Signaling) which were detected with secondary antibodies conjugated to Alexa 555 (1:1000, Invitrogen). Nuclei were visualized with DAPI. Images of single optical plane at the notochord were acquired with Leica SP5 confocal (Leica, Wetzlar, Germany). For quantification, the number of H3P-positive cells was counted manually in a region dorsal to the notochord, 500 µm from the tip of the tail.

## Statistical analysis

Data of at least three independent experiments are expressed as mean ± SD. Statistical analysis was determined using Students $t$-test using Excel (Microsoft, Redmond, WA, USA) or ANOVA followed by Post-hoc $t$-test and Bonferroni correction. For DNA injection experiment into zebrafish embryos, statistical significance was tested using chi-squared test.

## Acknowledgements

We thank Jana Petzold, Michaela Kümmel, and Tatyana Novoyatleva for their support; Gislene Pereira and Brian Polizzotti for discussions and critical reading of the manuscript; Sambra Redick and Stephen Doxsey for the RFP–Peri[CT] construct; Kunsoo Rhee for the *Pcnt B*-GFP construct.

## Additional information

### Funding

| Funder | Grant reference | Author |
|---|---|---|
| Alexander von Humboldt-Stiftung | Sofja Kovalevskaja Award | Felix B Engel |

| Funder | Grant reference | Author |
|---|---|---|
| Deutsche Forschungsgemeinschaft (DFG) | EN 453/9-1 | Felix B Engel |
| Friedrich-Alexander-Universität Erlangen-Nürnberg | EFI/ CYDER | Felix B Engel |
| European Society Of Cardiology (ESC) | German Cardiac Society (Klaus-Georg und Sigrid Hengstberger-Forschungsstipendium) | Gilbert Weidinger |

The funders had no role in study design, data collection and interpretation, or the decision to submit the work for publication.

## Author contributions

DCZ, Conception and design, Acquisition of data, Analysis and interpretation of data, Drafting or revising the article; SV, C-CW, Conception and design, Acquisition of data, Analysis and interpretation of data; TP, Acquisition of data, Analysis and interpretation of data, Contributed unpublished essential data or reagents; RB, ML, SH, FR, NF, Acquisition of data, Analysis and interpretation of data; AG, TB, Analysis and interpretation of data, Contributed unpublished essential data or reagents; SJ, Analysis and interpretation of data, Drafting or revising the article; GW, FBE, Conception and design, Analysis and interpretation of data, Drafting or revising the article

## Ethics

Animal experimentation: The investigation conforms with the Guide for the Care and Use of Laboratory Animals published by the Directive 2010/63/EU of the European Parliament and according to the regulations issued by the Committee for Animal Rights Protection of the State of Hessen (Regierungspraesidium Darmstadt) as well as Baden-Württemberg (Regierungspraesidium Tübingen). Extraction of organs and preparation of primary cell cultures were approved by the local Animal Ethics Committee in accordance to governmental and international guidelines on animal experimentation (protocol TS - 5/13 Nephropatho; Zebrafish protocol number o.183).

# Additional files

### Supplementary file

• Supplementary file 1. Homology analysis of PCNT to determine conserved regions for the identification of PCNT splice isoforms in the adult zebrafish heart.

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
