## [Decision Letter]

Thank you for sending your work entitled “Centrosome disassembly promotes a post-mitotic state in mammalian cardiomyocytes” for consideration at *eLife.* Your article has been favorably evaluated by Sean Morrison (Senior Editor and Reviewing Editor) and three reviewers, two of whom are members of our Board of Reviewing Editors.

The editor and the three reviewers discussed their comments before we reached this decision, and the editor has assembled the following comments to help you prepare a revised submission.

Zebrowski et al. show that centriole cohesion is lost as mammalian cardiomyocytes enter quiescence and propose that the disassembly of the centrosomes as MTOC is a key event to drive cardiomyocytes into the post-mitotic state. They further show that cardiomyocytes from fish and newt, which are known to maintain proliferative activity, maintain centrosome integrity, consistent with their model. The manuscript is well written, and the data are mostly convincing. Although the role of centrosome integrity in cell proliferation has been heavily studied, it has been always in the context of centrosome component mutants leading to a cell proliferation problem. In contrast, this manuscript provides a novel insight into how centrosome disassembly may be developmentally utilized to achieve cell cycle arrest. That conceptual distinction would be worth clarifying early in the manuscript.

1) All of the reviewers found the ideas in this manuscript conceptually interesting. The major concern is that all of the conclusions are based on correlations. Functional evidence is required to show that species-specific developmental changes in centrosome disassembly lead to differences in cardiomyocyte proliferation. The authors present data indicating that newt and zebrafish heart muscle cells contain centrosomes, whereas adult mouse heart cells do not. This result is exciting, as it could potentially explain how heart tissue can regenerate in newt and zebrafish. However, no further experiments are provided to test this hypothesis. It is not clear if centrosomes in adult newt and zebrafish cardiomyoctyes are indeed needed for regeneration. Nor is it clear if these centrosomes are even functional (i.e., do they nucleate MTs?). This is important to test, because tissue regeneration occurs in planarians even though they lack centrosomes. Furthermore, planarian regeneration proceeds normally even if centriole formation is abrogated via RNAi-mediated knockdown of SAS-4 (Azimzadeh, 2012, Science). Thus, the link between centrosome integrity and regenerative capacity requires more investigation. Does centrosome loss in cardiomyocytes eliminate the ability of adult newts or zebrafish to regenerate heart tissue?

2) In other systems, ciliogenesis is associated with G1/G0. Therefore, the lack of ciliogenesis in post-mitotic (supposedly G1/G0-arrested) cardiomyocytes requires further explanation/discussion. First, how firm are the data that support “G1/G0” arrest of cardiomyocytes? Is it possible that they are arrested in G2 phase of the cell cycle (thus do not form cilia)? Has anybody really examined the cell cycle status of post-mitotic cardiomyocytes? Ki67 is only a marker for “cycling cells” and does not tell in which cell cycle stage the cells are. Instead, true G1 markers such as Cip/Kip, Arf and DNA content (2N vs 4N) should be tested. Cardiomyocytes may represent a novel population of mammalian cells that enter quiescence and differentiate in G2 phase of the cell cycle (with 4C DNA content). This possibility is particularly plausible. Given the result shown in Figure 3 (p38i) are these cells in G2 with four centrioles (G2 cells have two centrosomes, each of which contains two centrioles)? Upon differentiation, those two centrioles would split as is shown in Figure 1, yielding 4 spindle poles when forced to enter mitosis due to p38i.

3) The authors show that the splicing isoform of Pericentrin (B and S) might underlie centrosome disassembly during cardiomyocyte cell cycle arrest. Two questions regarding this point: 1) does overexpression of Pericentrin B prevent cardiomyocytes from entering cell cycle arrest (possibly with a differentiation defect)? 2) Do fish and newts have the S isoform? Since most of the data in this paper are correlational, the sufficiency of the Pericentrin B form to inhibit cell cycle arrest should be shown. We understand that Pericentrin might not be the only regulator of centrosome integrity and the negative results would not necessarily be conclusive.

[Editors' note: further revisions were requested prior to acceptance, as described below.]

Thank you for resubmitting your work entitled “Centrosome disassembly promotes a post-mitotic state in mammalian cardiomyocytes” for further consideration at *eLife*. Your revised article has been favorably evaluated by Sean Morrison (Senior Editor), a Reviewing Editor, and three reviewers. The manuscript has been improved but there are some remaining issues that need to be addressed before acceptance, as outlined below:

The reviewers agreed that the manuscript was considerably improved. The remaining concern is that there is not yet direct evidence that induced centrosome disassembly impairs heart regeneration, although the conclusion is supported by ample indirect evidence. We understand that a causative relationship is often difficult to prove, and thus we would like to suggest toning down statements throughout the manuscript to make this point clear. For example, we would recommend changing the Abstract to state “…as well as a potential explanation for why zebrafish and newt, but not mammals, can regenerate their heart.”

Additionally there are still a few minor comments to be addressed prior to the acceptance of the manuscript.

Minor comments:

Figure 2, nonMC: The MT organization from the centrosome is unclear (because the nuclear envelope is not marked). Could the authors provide a better image to convey the point?

Figure 3: RFP-Peri^CT^ localization is somewhat misleading (additional markers for centrosome and nuclear envelope will help).

Figure 3: Scale bars are missing.

Figure 4—figure supplement 1: Change the color for “area of interest”. It is impossible to see any outlined area in the image.

Figure 2: Mono vs bi-nucleated cell. It's not very convincing that this is a bi-nucleated cell.

Are the cells in Figure 3 dividing?

---

## [Author Response]

*The manuscript is well written, and the data are mostly convincing. Although the role of centrosome integrity in cell proliferation has been heavily studied, it has been always in the context of centrosome component mutants leading to a cell proliferation problem. In contrast, this manuscript provides a novel insight into how centrosome disassembly may be developmentally utilized to achieve cell cycle arrest. That conceptual distinction would be worth clarifying early in the manuscript*.

Thank you very much for pointing out this issue. We have modified the Abstract and the Introduction to clarify this *conceptual distinction* early in the manuscript

*1) All of the reviewers found the ideas in this manuscript conceptually interesting. The major concern is that all of the conclusions are based on correlations. Functional evidence is required to show that species-specific developmental changes in centrosome disassembly lead to differences in cardiomyocyte proliferation. The authors present data indicating that newt and zebrafish heart muscle cells contain centrosomes, whereas adult mouse heart cells do not. This result is exciting, as it could potentially explain how heart tissue can regenerate in newt and zebrafish. However, no further experiments are provided to test this hypothesis. It is not clear if centrosomes in adult newt and zebrafish cardiomyoctyes are indeed needed for regeneration*.

While we have not provided experimental evidence that zebrafish require a centrosome for cardiac regeneration, we would like to emphasize the following points:

A) Our data clearly demonstrate that centrosome integrity is lost in mammalian cardiomyocytes in contrast to zebrafish and newt (see also B) below). As described in the Introduction, mammalian cell types require a centrosome for cell cycle entry and thus also for cardiac regeneration based on cardiomyocyte proliferation. In this regard, we have shown that “siRNA-mediated knockdown of Pericentrin, or overexpression of a RFP-tagged dominant negative C-terminal Pericentrin construct (RFP-Peri^CT^), which disrupts endogenous Pericentrin localization at the centrosome, suppressed P0-isolated cardiomyocyte proliferative potential (Figure 3).” Thus mammalian cardiomyocytes have to overcome a hurdle to enter the cell cycle that does not exist in zebrafish or newt cardiomyocytes.

B) The literature provides evidence that a functional centrosome is required for zebrafish development and cell cycle progression. For instance, plk4 depletion in zebrafish impairs centriolar biogenesis during development and increases premature cell cycle exit (independent of ciliogenesis defects) resulting in reduced zebrafish size (Martin et al., Nat Genetics, 2014). Other examples of centriole-associated genes whose depletion impairs zebrafish development exhibiting cell cycle defects resulting in reduced zebrafish size are “SCL/TAL1 interrupting locus (SIL or STIL)” (Pfaff et al., Mol Cell Biol 2007; Vulbrecht et al., J Cell Sci 2012; Sun et al., J Biol Chem, 2014) and “Centrin2” (Delaval et al., Cell Cycle, 2011). To clarify this issue we have modified the Discussion section.

In addition, we provide in collaboration with Gilbert Weidinger (Ulm) dnPCNT overexpression (RFP-Peri^CT^) data in zebrafish (delayed development and decresead mitosis index in RFP-Peri^CT^ RNA-injected embryos, reduced clonal size of RFP-Peri^CT^-positive cells in DNA-injected embryos) suggesting that also PCNT is required for cell proliferation in vivo. To clarify this issue we have modified the Results, Figure 4—figure supplement 1), Discussion and Methods sections accordingly.

C) The fact that vertebrate regeneration recapitulates development and that a centrosome is required for proliferation strongly suggests that cardiac regeneration in zebrafish (which occurs by cardiomyocyte proliferation) requires a functional centrosome. While we agree that there is currently no conclusive evidence for this, there is data that indirectly suggest that centrosome function might be required for cardiac regeneration in zebrafish. For example it has been shown that inhibition of Mps1 and plk1, factors implicated in centrosome assembly and maturation ([43], Cell Div; [30], J Cell Biol; [25], Moll Cell), impair cardiac regeneration in zebrafish after apex resection ([44], Science; Jopling et al., Nature, 2010). We have included this information in the revised manuscript in the Discussion section.

D) We agree that despite the developmental data it will be still interesting to determine the functional requirement of centrosomes in cardiac regeneration in zebrafish. Therefore, we work on establishing transgenic animals in which we can specifically induce in cardiomyocytes centrosome dysfunction. However, the generation of such a model alone takes at least 1 year. Thus, this is beyond the scope of this manuscript.

Nor is it clear if these centrosomes are even functional (i.e., do they nucleate MTs?).

To address this issue we have initiated collaborations with the groups of Steffen Just (Ulm) and Thomas Braun (MPIHL) to study centrosome function in adult primary zebrafish as well as newt cardiomyocytes. Our data based on immunofluorescence stainings and MTOC regrowth assays demonstrate that centrosomes in adult zebrafish as well as newt cardiomyocytes “nucleate MTs”. These results have been inserted in the Results section, Figure 4 and Figure 4—figure supplement 1, and the Methods section was changed accordingly.

*This is important to test, because tissue regeneration occurs in planarians even though they lack centrosomes. Furthermore, planarian regeneration proceeds normally even if centriole formation is abrogated via RNAi-mediated knockdown of SAS-4 (Azimzadeh, 2012, Science). Thus, the link between centrosome integrity and regenerative capacity requires more investigation*. *Does centrosome loss in cardiomyocytes eliminate the ability of adult newts or zebrafish to regenerate heart tissue?*

The phenomenon that planarians can regenerate in the absence of centrosomes is interesting but due to significant evolutionary and functional differences in cell cycle control we believe planarians cannot be compared to zebrafish or mammals. For example, although the planarian genome contains all components of the Rb pathway, they have undergone gene loss from the ancestral state, similar to other species in their phylum. A single Rb homolog (Smed-Rb) controls stem cell maintenance, similar to the Rb-homologs p107 and p130 in vertebrates. The most severe reduction occurred in E2F genes with a single remaining repressive E2F, in contrast to several repressive and activating E2Fs in mammals and zebrafish (Zhu et al., 2013, Dev Biol). In addition, genome and transcriptome analyses have revealed that planarians do not have either Cyclin E or Cyclin A homologs and further, cyclin-dependent kinase CDK2 (to which Cyclin E/A usually binds) is expressed at such low levels in planarians it is considered functionally dead (Zhu, 2013, Developmental Biology). This is important, as in mammals centrosome localization of Cyclin E and Cyclin A is required for G1/S cell cycle progression ([40], Cell Cycle). Thus, at least in the case of Cyclin E and Cyclin A, planarians lack centrosome-regulated cell cycle factors. Thus, the fact that planaria do not need a centrosome might not necessarily indicate that it is not required for proliferation in cell types of other species. To clarify this issue we have modified the Discussion section.

In addition, we would like to point out that regeneration in planarians occurs by neoblast (stem-like cells) proliferation while we study a well differentiated cell type, cardiomyocytes. Thus, the cell biology of the cells involved in planarian regeneration compared to zebrafish and mammalian cardiac regeneration is vastly different.

*2) In other systems, ciliogenesis is associated with G1/G0. Therefore, the lack of ciliogenesis in post-mitotic (supposedly G1/G0-arrested) cardiomyocytes requires further explanation/discussion. First, how firm are the data that support* “*G1/G0*” *arrest of cardiomyocytes? Is it possible that they are arrested in G2 phase of the cell cycle (thus do not form cilia)? Has anybody really examined the cell cycle status of post-mitotic cardiomyocytes? Ki67 is only a marker for* “*cycling cells*” *and does not tell in which cell cycle stage the cells are. Instead, true G1 markers such as Cip/Kip, Arf and DNA content (2N vs 4N) should be tested. Cardiomyocytes may represent a novel population of mammalian cells that enter quiescence and differentiate in G2 phase of the cell cycle (with 4C DNA content)*.

There is a large consensus of information that neonatal cardiomyocyte nuclei as well as the majority of adult cardiomyocytes of most mammals including rat and mice are diploid – indicating that they are in a G1/G0 state (e.g. Brodsky and Uryvaeva, 1977, Int Rev Cytol; Adler et al., 1996, Virchows Arch).

In addition, it has been shown that the cell cycle arrest of cardiomyocytes is induced by developmentally regulated upregulation of p21 and p27 expression (Horky et al., 1997, Physiol Res; Poolman et al., 1998, Int J Cardiol; Koh et al., 1998, J Mol Cell Cardiol; Burton et al., 1999, Dev Biol; Tane et al., 2014, Biochem Biophys Res Commun).

That the majority of rat cardiomyocytes are arrested in G1 phase has also been demonstrated by FACS analysis (e.g. Poolman and Brooks, 1998, J Mol Cell Cardiol; Poolman et al., 1998, Int J Cardiol; Engel et al., 1999, Circ Res; Ebelt H. et al., 2005, Circ Res; Liu et al., 2010, Cir Res).

To clarify this issue we have changed the text in the Results section. In addition, to further substantiate that the cardiomyocytes used in our study are arrested in G1/G0, we have isolated P0/1 and P3 cardiomyocytes, cultured them for 3 days in the absence of mitogenic stimuli, as done with ciliogenesis assay, and then determined their cell cycle stage by FACS. As shown in Figure 2—figure supplement 1, the vast majority of P0/1- and P3-isolated cardiomyocytes are in G0/G1-phase (Figure 2—figure supplement 1).

Therefore, the inability for postnatal cardiomyocytes to perform ciliogenesis is not due to being arrested at the G2 phase of the cell cycle.

*This possibility is particularly plausible. Given the result shown in*
Figure 3
*(p38i) are these cells in G2 with four centrioles (G2 cells have two centrosomes, each of which contains two centrioles)? Upon differentiation, those two centrioles would split as is shown in*
Figure 1*, yielding 4 spindle poles when forced to enter mitosis due to p38i*.

Figure 3 shows histone H3-positive cardiomyocytes with condensed chromosomes in metaphase. Thus, these cells are not in G2 phase but in mitosis.

3) The authors show that the splicing isoform of Pericentrin (B and S) might underlie centrosome disassembly during cardiomyocyte cell cycle arrest. Two questions regarding this point: 1) does overexpression of Pericentrin B prevent cardiomyocytes from entering cell cycle arrest (possibly with a differentiation defect)?

Unfortunately, due to technical issues it is not possible to answer this question:

1) The standard technique (and by far the most efficient) to ectopically express proteins in cardiomyocytes is the infection with adenoviruses. Unfortunately, the gene encoding Pericentrin B is too big to generate an adenovirus.

2) Transfection technologies like Lipofectamine are not efficient in cardiomyocytes. Electroporation provides an alternative but its efficiency decreases markedly with the construct size.

3) The only assay that came to our mind to test whether “overexpression of Pericentrin B prevent cardiomyocytes from entering cell cycle arrest” is to culture fetal cardiomyocytes in serum. However, it takes nearly a week before these cells exit the cell cycle and thus would require long-term overexpression.

Nevertheless, we performed Pericentrin B overexpression experiments. While transfection efficiencies is low, it still allowed us to evaluate cellular localization. Our data indicate that PCM1 is lost from the centrosome before Pericentrin B. As PCM1 is required for the recruitment of centrosome proteins such as Pericentrin B to the centrosome it is unlikely that Pericentrin B overexpression can prevent cardiomyocytes from entering cell cycle arrest. To verify this hypothesis we have overexpressed Pericentrin B in non-myocytes and cardiomyocytes and analyzed its localization. While Pericentrin B-GFP localized to the centrosomes of non-myocytes and most E15-cardiomyocytes, it failed to localize to centrioles in the majority of P0 and P3 cardiomyocytes. These observations are consistent with the absence of PCM1 at the centrosome post birth. We have added these data to the revised manuscript and Figure 1—figure supplement 2.

2) Do fish and newts have the S isoform?

Unfortunately, there are no antibodies available for Pericentrin in zebrafish or newt. Thus, we cannot determine whether Pericentrin S is expressed on protein level. Also, the mRNA sequence encoding the Pericentrin S isoform is neither known for zebrafish nor newt. For the newt no sequence data are available for the Pericentrin gene. However, for zebrafish the full-length gene encoding Pericentrin is known.

To determine if zebrafish might express the Pericentrin S isoform we performed RT-PCR experiments. A particularity of mouse and rat Pericentrin S is the inclusion of a unique exon. Thus we have aligned the sequences of these exons with the corresponding regions of the zebrafish and human Pericentrin B gene, utilizing the online software Clustal W2. This analysis indicated a longer stretch exhibiting some sequence conservation (see [Supplementary-material SD1-data]).

We designed a forward primer (marked in yellow) in this region and were able to amplify a product utilizing a reverse primer in exon 25 (5′ CAGCGCATCGTCAAAGAGATG3′, spanning intron 24, 255 bp). Our analyses indicate that heart muscle expresses a putative Pericentrin S, yet at a much lower level than total Pericentrin (forward: 5′ CAAGCGCATGGAGCAGGTGAAG 3′, reverse: 5′ CCTGCTCTGAGGACTGAAGT GCTGA 5′; see Figure 5 below).

Author response image 1.RNA was obtained from adult zebrafish hearts. RT-PCRs were performed to amplify total Pericentrin (should amplify all known isoforms, total PCNT) or only the PCNT S isoform. As control genomic DNA was utilized (gDNA). Yellow star: indicates putative PCNT S.**DOI:**
http://dx.doi.org/10.7554/eLife.05563.015

However, whether this isoform is indeed Pericentrin S, whether it is expressed at protein level or whether it plays the same role in zebrafish cardiomyocytes as observed in mammalian cardiomyocytes remains unclear. Thus, we have not included these data in the manuscript.

[Editors' note: further revisions were requested prior to acceptance, as described below.]

*The reviewers agreed that the manuscript was considerably improved. The remaining concern is that there is not yet direct evidence that induced centrosome disassembly impairs heart regeneration, although the conclusion is supported by ample indirect evidence. We understand that a causative relationship is often difficult to prove, and thus we would like to suggest toning down statements throughout the manuscript to make this point clear. For example, we would recommend changing the Abstract to state* “*…as well as a potential explanation for why zebrafish and newt, but not mammals, can regenerate their heart*.”

We agree that we have not proven that induced centrosome disassembly impairs heart regeneration. To make this point clear we have made adjustments throughout the text.

*Additionally there are still a few minor comments to be addressed prior to the acceptance of the manuscript*.

Minor comments:

Figure 2*, nonMC: The MT organization from the centrosome is unclear (because the nuclear envelope is not marked)*. *Could the authors provide a better image to convey the point?*

To address this we have added new regrowth assay images using DAPI to visualize the nucleus in non-myocytes (Figure 2—figure supplement 1). Here, one can clearly see that microtubules emanate from the centrosome and not from the nuclear envelope in non-myocytes (which is also indicated in Figure 2; note yellow arrowheads). The Results section was changed accordingly.

Figure 3*: RFP-Peri*^*CT*^
*localization is somewhat misleading (additional markers for centrosome and nuclear envelope will help)*.

It has already previously been demonstrated that RFP-Peri^CT^ localizes to the centrioles ([36] and [20]). To verify this, we have provided in Figure 3—figure supplement 1 representative images of P0-isolated cardiomyocytes transfected with RFP-Peri^CT^ stained for cardiomyocytes (Troponin I) and centrioles (γ-tubulin). Nuclei were visualized with DAPI. We have accordingly added a new figure legend and altered the Results section as follows: “Subsequently, we tested whether centrosome […] P0-isolated cardiomyocyte proliferative potential (Figure 3).”

Figure 3*: Scale bars are missing*.

The scale bars have been added as requested.

Figure 4—figure supplement 1*: Change the color for* “*area of interest*”*. It is impossible to see any outlined area in the image*.

To clarify this issue we have replaced the current brownish outline of the “area of interest” with a thicker yellow outline.

Figure 2*: Mono vs bi-nucleated cell*. *It's not very convincing that this is a bi-nucleated cell.*

In order to clarify this issue we have decided to show the same images at a lower magnification. The nuclei become visible as dark spots in the Troponin I-stained cardiomyocytes. To further clarify this issue we have indicated the nuclei with a white dotted line.

*Are the cells in*
Figure 3
*dividing?*

We have published in 2005 (Engel et al., Genes & Development) that the treatment of adult rat cardiomyocytes with FGF1 and an inhibitor of p38 MAP kinase (p38i) induces cell cycle re-entry, mitosis, and cytokinesis. Based on our data FGF1/p38i induces in a subset of adult cardiomyocytes cell division. Yet, from a still photo like in Figure 3, it is impossible to say if this particular cell will divide or not. Thus, we stated in the manuscript only that: “Similarly, adult-isolated cardiomyocytes induced to re-enter the cell cycle by p38i + FGF1 also exhibited multiple γ-tubulin foci during metaphase (Figure 3).”